# Assessment of Barriers and Challenges to Screening, Diagnosis, and Biomarker Testing in Early-Stage Lung Cancer

**DOI:** 10.3390/cancers15051595

**Published:** 2023-03-03

**Authors:** Reza Zarinshenas, Arya Amini, Isa Mambetsariev, Tariq Abuali, Jeremy Fricke, Colton Ladbury, Ravi Salgia

**Affiliations:** 1Department of Radiation Oncology, City of Hope National Medical Center, Duarte, CA 91010, USA; 2Department of Medical Oncology, City of Hope National Medical Center, Duarte, CA 91010, USA

**Keywords:** lung cancer, screening, biomarkers, LDCT, COVID-19

## Abstract

**Simple Summary:**

Lung cancer management continues to evolve with improvements in survival across all stages. The review highlights the current data supporting screening and discusses barriers to screening and potential opportunities to improve screening access. Further, the review discusses the current challenges in diagnosis and biomarker testing in early stage lung cancer and ways in which these can be improved upon.

**Abstract:**

Management of lung cancer has transformed over the past decade and is no longer considered a singular disease as it now has multiple sub-classifications based on molecular markers. The current treatment paradigm requires a multidisciplinary approach. One of the most important facets of lung cancer outcomes however relies on early detection. Early detection has become crucial, and recent effects have shown success in lung cancer screening programs and early detection. In this narrative review, we evaluate low-dose computed tomography (LDCT) screening and how this screening modality may be underutilized. The barriers to broader implementation of LDCT screening is also explored as well as approaches to address these barriers. Current developments in diagnosis, biomarkers, and molecular testing in early-stage lung cancer are evaluated as well. Improving approaches to screening and early detection can ultimately lead to improved outcomes for patients with lung cancer.

## 1. Introduction

Lung cancer has a low five-year survival rate due in part to delays in diagnosis and presentation of advanced disease, making early detection critical [1]. In the US, less than 20% of those diagnosed with lung cancer presented with localized disease [2]. The 2011 National Lung Screening Trial (NLST) reported a relative reduction of mortality from lung cancer of 20.0% with low-dose computed tomography (LDCT) screening in comparison to chest radiography [1,3]. The European Nederlands Leuvens Longkanker Screenings Onderzoek (NELSON) trial showed LDCT screening for high-risk patients led to a 26% reduction in lung cancer mortality. The Multicentric Italian Lung Detection (MILD) trial demonstrated a 39% reduction in lung cancer mortality at the 10-year mark as a result of screening over five years [4]. After the NLST’s publication, the United States Preventative Task Force (USPSTF), American Cancer Society, and National Comprehensive Cancer Network (NCCN) developed recommendations for annual LDCT screening for high-risk patients [1]. The USPSTF currently supports annual screening for lung cancer with LDCT for adults aged 50 to 80 years who have a 20 pack-year smoking history and currently smoke or have quit within the past 15 years (B recommendation) [5]. The American Academy of Family Physicians reports a B recommendation for the same population [6]. Current NCCN guidelines recommend annual LDCT for individuals 50–80 years old with greater than or equal to a 20 pack-year history of smoking [7]. Despite these recommendations, LDCT screening has been underutilized [8]. The goal of this narrative review is to evaluate and address the potential barriers to screening and diagnosis as well as to explore biomarkers and molecular testing approaches in early-stage non-small cell lung cancer (NSCLC).

## 2. Barriers to Screening

The percentage of eligible smokers who reported LDCT screening only increased from 3.3% in 2010 to 3.9% in 2015 (*p* = 0.60) according to respondents of the National Health Interview Survey in the United States [9]. Of note, in 2015 the Centers for Medicare and Medicaid Services (CMS) approved insurance coverage for LDCT for high-risk patients. In 2016, only 1.9% of eligible patients were screened [1]. Further, a Center for Disease Control analysis of 10 states in 2017 found that 12.5% of eligible candidates received LDCT in the 12 months prior [8]. There are multiple reasons why there is not greater incorporation of LDCT screening in the United States. Physician referral for LDCT is one primary barrier which is multifactorial and may be related to busy practice patterns, lack of awareness of screening guidelines, and patient-specific barriers discussed below. Other barriers to LDCT include lack of notifications from the electronic medical record (EMR), patient refusal, time constraints, varying provider knowledge about insurance coverage, cost concerns, concern for false positives and overtreatment, lack of patient knowledge of screening guidelines, lack of patient access to screening, concerns about radiation exposure, inadequate staffing, and inconsistent lung cancer screening recommendations across organizations [1,10].

Screening techniques have improved significantly in recent years. Regarding risks related to radiation exposure due to CT lung cancer screening (CTLS), the risk appears to be too low to measure. Further, there have been no definitive cases of radiation-induced malignancies that have been reported. However, in one study, it was estimated that 10 annual LDCTs produce a 0.26 to 0.81 lifetime risk of major cancers for every 1000 people screened [2]. Over-diagnosis and false-positive rates leading to additional workup do not appear to significantly impact lung cancer patients either and should not be a barrier for screening [11]. One analysis of the NLST trial demonstrated that 779 per 1000 people would have a normal screen and no diagnosis of lung cancer based on Lung-RADS criteria and 180 per 1000 people were found to have a false positive. Of the 180 false positives, 13 would require an invasive procedure to exclude lung cancer, and 1 per 2500 people would have a major complication from an invasive procedure. Further, they reported that 1 per 5000 screened would die within 60 days of the invasive procedure from any cause [12]. Secondary analysis of the Danish Lung Cancer Screening Trial revealed an over-diagnosis rate of 67.2% [13,14]. However, secondary analysis of the NLST showed that complications from invasive procedures were low overall [15]. Standardized screening systems could help reduce rates of over-diagnosis and complications [11].

Geography can be a barrier to screening (Table 1). In one study from the US, the South had the greatest eligible number of smokers and the most screening sites. Nonetheless, it had one of the lowest screening rates at 1.7%, compared with 2.1% in the Midwest, and 3.9% in the Northeast. The West had the lowest screening rate at 1.1% but also had the second fewest eligible smokers. Higher screening rates in the Northeast could partially be linked to having a higher density of physicians per capita, rate of insurance, per capita income, and education level. The South had the lowest rate of insured patients. Furthermore, there is a lower likelihood of having screening sites in rural areas outside of the Northeast [10]. In the US, 14.9% of patients did not have a lung cancer screening center within 30 miles, a problem particularly prevalent in rural areas (Table 1). The shortage of LDCT screening centers in rural regions is particularly concerning given the number of at-risk individuals in rural regions. Expanding telehealth coverage is one approach to address this challenge [16].

Health insurance status continues to factor into those who undergo screening in the US. While Medicare and most commercial insurers started covering LDCT starting in 2015, Medicaid (which covers 19% of the US population) may not always cover. Moreover, 9% of the population remains uninsured [10]. Greater than half the patients who are eligible for lung cancer screening according to the USPTF screening guidelines are either uninsured or have Medicaid insurance. Uninsured patients are 72% less likely to get screened compared with insured patients [17,18]. Medicaid coverage of lung cancer screening depends on whether the state adopts Medicaid expansion [17]. One approach to address this limitation would be to expand Medicaid coverage of lung cancer screening to all states [16]. Other approaches for improving LDCT include making lung cancer screening a Center for Medicare and Medicaid Services (CMS) national quality measure and increasing awareness and compliance among physicians and patients [10]. Another proposal to address screening disparities in underserved populations is to link American College of Radiology (ACR) screening sites with Federally Qualified Health Centers (FQHC) [19]. Race and ethnicity also impact rates of screening. One limitation of the NELSON and NLST studies was that their populations consisted mainly of white men. Nonetheless, minority groups are disproportionately impacted by lung cancer and are not well represented in studies [20]. In particular, African Americans have the highest rates of lung cancer mortality despite similar smoking rates compared with whites [19]. In the NLST trial, only 4% of participants were African American and yet a secondary analysis suggested that African Americans had a significant benefit from LDCT screening [21]. One approach to address disparities faced by African Americans and other at-risk populations, is to have local and state organizations partner to develop outreach programs, educational materials, and increase awareness. In addition, developing strategies to address stigma, clinician implicit bias, and nihilism is imperative [19]. There is evidence to suggest that individuals of a higher socioeconomic status and men are overrepresented in lung cancer screening programs. Finding approaches to engage women and individuals of lower socioeconomic statuses is critical [22].

Despite the underutilization of LDCT screening, efforts should be made to educate patients on both the risks and benefits of LDCT. The USPSTF recommends that shared decision-making be included in discussions regarding the pros and cons of LDCT screening. Shared decision-making is required for Medicare reimbursement, which could be a barrier given the shortage of time in visits [10]. Nonetheless, shared decision-making surrounding lung cancer screening is often underutilized, and the potential harms of screening are not sufficiently explained (should there be a comma between 22 and 23) [23,24].

Lung cancer screening in non-smokers remains challenging and to date there are no standardized screening guidelines for non-smokers [25]. Although lung cancer is strongly associated with smoking, the rate of lung cancer in non-smokers has been increasing in the US [26]. In one study using the United States (US), it was found that 12.5% of patients with lung cancer were never smokers. Women were more likely than men to be never smokers with lung cancer. From a race/ethnicity perspective, Asian/Pacific Islanders comprised the highest percentage of never smokers with lung cancer [26]. Risk factors for lung cancer in non-smokers include increased age, secondhand smoke, environmental exposures, radon exposure, genetic factors, underlying lung disease, and oncogenic viruses [25]. The racial discrepancies in lung cancer among non-smokers deserves further investigation. As one example of this, the majority of Han Chinese women with lung cancer have been non-smokers. Exposure to cooking oil fumes has been explored as one potential explanation for this discrepancy [27]. The upcoming FANS (Female Asian Never Smokers) Study will further explore potential causes of lung cancer in never-smoker Asian American women, which could in turn inform screening strategies for never smokers with lung cancer [28].

## 3. Screening in the Era of Coronavirus Disease 2019 (COVID-19)

In the initial stages of the COVID-19 pandemic, the American College of Chest Physicians recommended delaying initial and repeat annual lung cancer screening to avoid COVID-19 exposure in the general population [29,30]. One study attempted to measure the impact of COVID-19 on lung cancer screening through a prospectively maintained database recording patients in a lung cancer screening program when LDCT screening was temporarily suspended during the initial stages of the pandemic. The percentage of patients with lung nodules suspicious of malignancy when screening resumed increased to 29% from their 8% institution baseline levels, causing a sudden rise in referrals to thoracic surgery or interventional pulmonology [29]. Despite the challenges that the COVID-19 pandemic brought, the pandemic also led to an increased use of telehealth services which offered added convenience for the shared decision-making required for lung cancer screening [31]. One broad approach to help categorize and summarize the LDCT screening barriers discussed is to view them as barriers for providers, barriers for patients, screening and treatment concerns, healthcare disruptions, and screening limitations (Figure 1). From there a number of potential solutions can be applied to improve LDCT screening access including more screening centers, expanding insurance coverage for LDCT, identifying populations that have limited access to screening or have hesitations to undergo screening, and better education primary care physicians on the importance of LDCT for those who meet criteria for screening (Figure 2).

One area of advancement within LDCT is the integration of artificial intelligence (AI) for early detection of lung cancer. A review article, published in 2021, evaluated comparative studies between AI algorithms and humans. These studies, many of which were from over a decade ago, showed that AI algorithms were either a little worse or equivalent to their human counterparts. However, the false-positive rate for the AI algorithms was higher [32]. Newer approaches have been developed to reduce false positives. For example, computer-aided detection (CAD) systems for detecting pulmonary nodules is being evaluated as a second reader for radiologists. CAD systems have also been used to distinguish between six nodule types (perifissural, spiculated, solid, part-solid, non-solid, calcified), and they showed similar performance to a human expert when the deep learning (DL) system was trained on data from the Italian MILD screening trial and validated on data from the Danish LCS trial [33]. In one study, a deep learning model for the analysis of malignancy risk in lung cancer screening CTs was developed and applied to 6716 NLST cases for which a 94.4% area under the curve was achieved. This model outperformed six radiologists and had an 11% absolute reduction in false positives and a 5% reduction in false negatives when prior CT imaging was not available. The model’s performance was comparable to the same radiologists when prior imaging was available [34].

Furthermore, AI can be implemented to identify smoking-related diseases on LDCT, including coronary artery calcification and cardiovascular events, emphysema, and osteoporosis and fragility fractures. As one example of this expanded role, the AI-RAD Companion by Siemens Healthineers uses DL algorithms to provide assessments of emphysema and coronary artery calcification in addition to detecting lung nodules. DL algorithms that can assess for smoking-related diseases could broaden the impact of lung cancer screening [33].

## 4. Challenges in Treatment Diagnosis

One challenge for lung cancer diagnosis is tumor heterogeneity, which makes molecular analysis important [7]. The Clinical Lung Cancer Genome Project and Network Genomic Medicine evaluated 1255 lung tumors and found that 55% had at minimum one genomic alteration that is possibly amenable to targeted therapy. Diagnostic platforms available for genomic profiling of lung cancers have increased in recent years given the importance of genomic testing before initiating treatment for advanced lung cancer patients. Barriers to broader adoption of molecular testing globally include availability of tissue, cost of molecular testing, quality and standards, access to molecular testing within their institution, awareness of the most recent guidelines, turnaround time for results, and inconclusive results [35].

Appropriate staging may be another barrier in treatment diagnosis. CT imaging is the most common imaging modality for the staging of lung cancer. However, CT of the thorax only has a sensitivity of 55% and specificity of 81% in detecting malignant mediastinal lymph nodes. PET-CT offers higher accuracy (sensitivity 80% and specificity of 88%) for detecting metastasis to the mediastinal lymph nodes. However, PET-CT can face challenges differentiating between inflammation, infection, and malignant disease, which raises the importance of tissue sampling. PET-CT can also yield a high false-negative rate in situations in which lymph nodes are moderately enlarged or not at all [36]. Therefore, in particular for mediastinal/hilar staging, mediastinoscopy or EBUS are recommended, and a PET-CT should not obviate the need for these procedures.

Additionally, PET radiotracers are currently being evaluated that may be more cancer-specific than the current fluorodeoxyglucose (FDG) that is used. For example, fibroblast activation protein inhibitor (FAPI), which is expressed in the vast majority of epithelial cancers, has been investigated in the setting of lung cancer imaging. The fibroblast activation protein inhibitor (FAPI), binds to FAP. FAPI derivatives can be radiolabeled with Gallium-68 [^68^Ga]. For detecting of lung metastasis, there is evidence to suggest that [^68^Ga]Ga-FAPI PET/CT offers higher maximum standardized uptake value and tumor-to-background ratio compared with [^18^F]FDG PET/CT making this modality potentially useful for staging. Furthermore, prognostic applications of [^68^Ga]Ga-FAPI PET are being explored [37].

## 5. Biomarkers/Molecular Testing

### 5.1. Current Testing Paradigm and Barriers

Currently molecular biomarker testing is standard for advanced and metastatic NSCLC as it impacts selection of systemic therapy. Broad panel-based Next Generation Sequencing (NGS) is usually recommended for patients as it includes the most common molecular alterations with specific therapy options. If the results of the broad panel-based NGS approach are unrevealing, then a ribonucleic acid (RNA)-based NGS is recommended. Other techniques that can be employed are real-time PCR (polymerase chain reaction), Sanger sequencing, fluorescence in situ hybridization, and Immunohistochemistry (IHC). ALK (Anaplastic Lymphoma Kinase) rearrangements can be detected via IHC or fluorescence in situ hybridization (FISH). ROS1 rearrangements are screened with IHC and validated with FISH. BRAF (B-Raf proto-oncogene) point mutations and KRAS (KRAS proto-oncogene) point mutations can be detected with real-time PCR, Sanger sequencing, and NGS. NGS is recommended for the detection of MET (mesenchymal–epithelial transition) exon 14 (METex14) skipping variants. FISH can be used to detect RET (rearranged during transfection) Gene Rearrangements. NTRK1/2/3 (neurotrophic tyrosine receptor kinase) gene fusions can be detected through NGS, FISH, IHC, and PCR [7]. IHC for PD-L1 (Programmed death ligand 1) can be implemented to find a disease that is likely to respond to first-line anti-PD-1/PD-L1 and can be viewed as a predictive biomarker [20]. Plasma cell-free/circulating tumor DNA (ctDNA) has high specificity but compromised sensitivity. For this reason, ctDNA should not replace histologic tissue diagnosis [7]. Current guidelines from the College of American Pathologists, International Association for the Study of Lung Cancer (IASLC), and the Association of Molecular Pathology recommend testing for EGFR (Epidermal Growth Factor Receptor) mutations, ALK rearrangements, ROS1 rearrangements, BRAF, RET rearrangements, and MET exon 14 skipping mutations for newly diagnosed lung adenocarcinoma [20].

### 5.2. Liquid Biopsy

Liquid biopsy is an alternative form of biopsy which complements current tissue biopsy. A liquid biopsy is an analysis of any tumor-derived product in the blood or serum. Compared with tissue biopsies, they assess the spatial and temporal heterogeneity of the tumor and may be a useful option when tissue is limited or unavailable [38]. Analyzing ctDNA in patients with NSCLC is a common application of liquid biopsy and can offer similar utility as tissue biopsy in many settings [39]. Liquid biopsies can help alleviate challenges and delays in getting additional tissue often needed for molecular analysis. Liquid biopsy assays have been studied in the EGFR population. One meta-analysis comparing EGFR mutations in plasma and tumor tissue found a sensitivity of 67% and specificity of 94% for plasma measurements [40]. While there are US Food and Drug Administration (FDA) approved tests for EGFR mutations (cobas EGFR Mutation Test v2), NGS allows the ability to analyze multiple genes at the same time or the entire cancer genome [41].

There are a number of studies evaluating the role of ctDNA. The Galleri multicancer early detection (MCED) test identifies circulating cell-free DNA (cfDNA) in blood using next-generation sequencing to detect DNA methylation for early detection of cancer as well as identifying the organ of origin for the cancer. The Circulating Cell-free Genome Atlas (CCGA), STRIVE, SUMMIT, and PATHFINDER studies are four trials using this technology, with the CCGA being the initial development of the test [42]. The CCGA was used to validate an MCED test using cfDNA with machine learning to detect cancer signals with a single blood draw. This study was divided into three substudies. In the first substudy, whole genome methylation was identified as the method to be used for further development. The second study supported further cfDNA sequencing evaluation in a prospective population-level study [43]. In the third study, a pre-specified subset analysis of 4077 participants (2823 with cancer and 1254 non-cancer) was used for clinical validation of this test [44]. The STRIVE trial has a goal of validating the Galleri test for early detection of cancer. The SUMMIT trial has the goal of validating the Galleri test by measuring cancer incidence. The PATHFINDER evaluates implementation of the test in clinical practice [42].

The combination of blood testing (liquid biopsies) with imaging has been investigated as a screening option for cancer. The DETECT-A (Detecting cancers Earlier Through Elective mutations-based blood Collection and Testing) trial consisted of a blood test designed to detect DNA mutations and protein biomarkers in a cohort of 10,006 women from the ages 65 to 75 years of age [45]. Positive blood tests in this study were followed by a PET-CT scan. Of the 96 cancers detected in this cohort, 26 were first detected by a blood test (of which nine were lung cancers). Twenty-four cancers (of which three were lung cancers) were detected through standard-of-care screening [45].

In 2020, the US FDA approved Guardant 360 and FoundationOne Liquid CDx as ctDNA assays to detect genomic alterations in patients with advanced-stage solid malignancies and are commonly used at many institutions. Another avenue being explored is the use of ctDNA for detecting minimal residual disease (MRD). In a study published in 2017, it was found that ctDNA detected after curative-intent NSCLC treatment could predict relapse often prior to radiologic evidence of relapse. In 2020, updated results were presented, and it was determined that among those with disease recurrence, 82% of patients had ctDNA that was detected at or before clinical relapse [46].

### 5.3. Future Advancements in Biomarker Technologies

Some other example biomarkers include autoantibodies, complement fragments, microRNA, methylated DNA, and proteins [47]. Autoantibodies are in response to tumor antigens and can appear even in the preclinical phase. They have the benefit of being specific but not sensitive [48]. Lung cancer can also activate the classic complement pathway [48]. Studies have suggested complement factors, such as factors H, C5a, and C4d as lung cancer biomarkers in lung cancer cell lines [47]. Like autoantibodies, complement fragments suffer from high specificity and low sensitivity [48]. Circulating microRNA, which has been explored for cancer diagnosis and prognosis, has been shown to reduce LDCT false-positive rates in two large retrospective studies. Serum antigens have also been explored to improve diagnostic accuracy. Liquid chromatography–mass spectrometry (LC–MS) has been explored for biomarker research in lung cancer. In one case-control study involving 24 bronchoalveolar lavage extracellular vesicle samples, state-of-the-art LC–MS applied to bronchoalveolar lavage small vesicles found that there is evidence to suggest proteome complexity is correlated with stage 4 lung cancer and mortality [49]. Furthermore, in this study a possible therapeutic target, Cytosine-5-methyltransferase 3β (DNMT3B) complex protein, which is an epigenetic modifier, was found to be upregulated in tumor tissues and bronchoalveolar lavage extracellular vesicles [49]. Additionally, in a real-life cohort study of 97 patients, LC–MS was used to identify 34 proteins in pleural effusions that are associated with survival. The proteins identified could inform research into more personalized therapeutic options [50]. Other future directions for biomarkers include exhaled breath (EB) biomarkers, sputum cell-based image analysis, metabolomics, genomic predispositions, radiomics, and artificial intelligence [48].

Biosensors are also being explored as an avenue for lung cancer diagnoses [51]. Optical, electro-conductive, piezoelectric, and amperometric biosensors are types of biosensors. In the realm of optical biosensors: surface plasmon resonance (SPR)-based sensors, SPR-based sensor-on-chip, and quantum dot-based sensors are some examples [47].

### 5.4. Additional Barriers to Biomarker Testing

Despite the rapid growth of biomarker testing, many challenges remain [52]. For example, biopsy tissue samples are often inadequate for biomarker testing, or the tests themselves may face technical problems. Coordination with services such as pulmonology and interventional radiology can help, and taking steps to improve tissue handling can be taken as well. Additionally, as discussed earlier, blood biomarker testing may complement tissue testing and potentially one day obviate the need for tissue or, at a minimum, repeat biopsies for additional tissue for molecular testing [53].

Access to personalized medicine can be more challenging in the community setting. A cloud-based virtual molecular tumor board (VMTB) can be an effective way to establish a relationship between community oncologists and academic site physicians [54]. Furthermore, the molecular tumor board (MTB) model can be utilized at the community level, either through virtual or physical collaborations, and can be especially useful in settings such as the current COVID-19 pandemic. Furthermore, the use of vendor-based oncology clinical pathways can be one way to guide physicians in decision-making regarding a patient case amidst the rapidly changing guidelines in oncology. Another way to optimize the delivery of care in the community setting is to use a standardized electronic health record system in both the academic and community settings [55] (Figure 2).

## 6. Conclusions

In this review, we explore lung cancer screening with LDCT and the advantages and potential disadvantages of LDCT screening. Barriers to wider implementation of LDCT screening were investigated, including socioeconomic, racial, geographic factors, and healthcare interruptions, and potential strategies to try to reduce these barriers were discussed (Figure 2). Furthermore, current paradigms in lung cancer biomarkers were also discussed, as well as the future avenues and opportunities with lung cancer biomarkers. Finally, potential approaches to encourage wider adoption of personalized medicine approaches were discussed. As the field validates the current use of biomarkers and incorporates radiomic features found on imaging, including those from LDCT scans, tools can be developed to better detect early-stage NSCLC. AI may play a major role in storing and interpreting the large amounts of molecular and digital data to aid clinicians in improving early detection of lung cancer. In conclusion, in the current era of early-stage detection, while LDCT plays a pivotal role, the additional biomarkers discussed could further stratify low- and high-risk patients, leading to quicker diagnoses and potential treatment options.

## Figures and Tables

**Figure 1 cancers-15-01595-f001:**
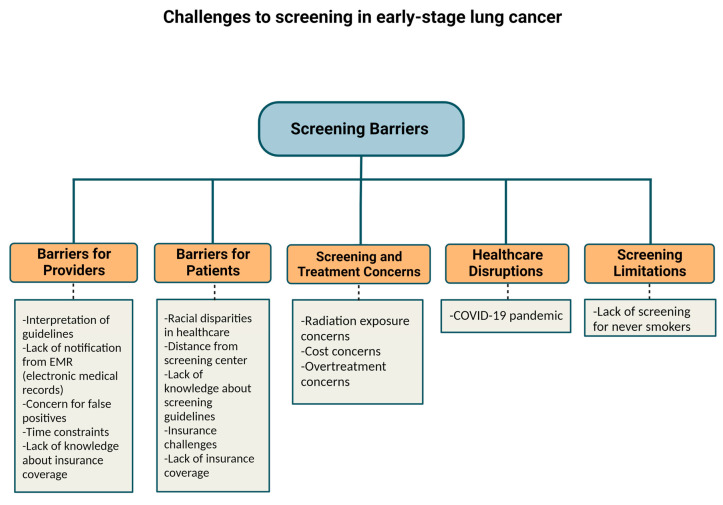
Low-dose computerized tomography (LDCT) screening barriers include barriers for providers, barriers for patients, screening and treatment concerns, healthcare disruptions, and screening limitations. Identifying these barriers is a critical step towards wider adoption of LDCT screening for at-risk patients. Created with BioRender.com accessed on 3 February 2023.

**Figure 2 cancers-15-01595-f002:**
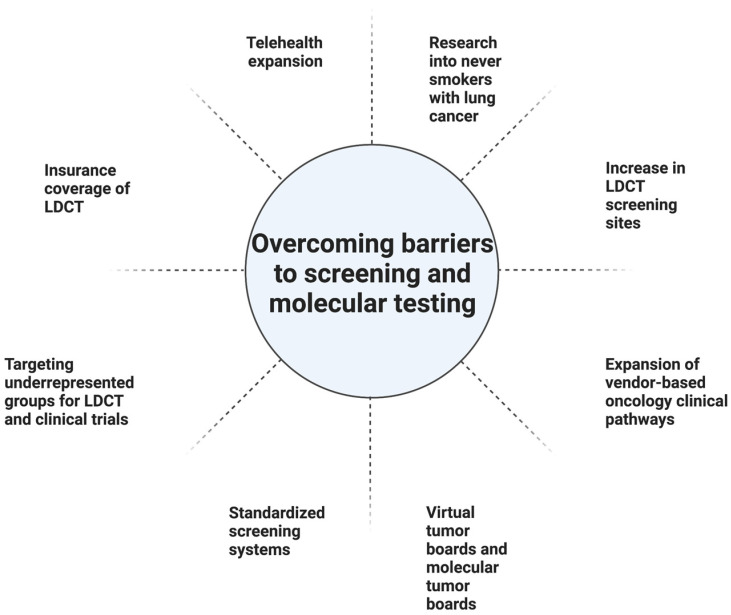
Listed is a summary of strategies that can be implemented to overcome barriers to screening and molecular testing. Created with BioRender.com accessed on 3 February 2023.

**Table 1 cancers-15-01595-t001:** Rates of Screening by Region in the United States.

Total Percentage Screened in US in 2016	2.0% 9
Percentage screened in West in 2016	1.1% (Region ranked 3rd out of 4 regions in terms of number of eligible smokers for screening) 9
Percentage screened in South in 2016	1.7% (Region ranked 1st out of 4 regions in terms of number of eligible smokers for screening) 9
Percentage screened in Midwest in 2016	2.1% (Region ranked 2nd out of 4 regions in terms of number of eligible smokers for screening) 9
Percentage screened in Northeast in 2016	3.9% (Region ranked 4th out of 4 regions in terms of number of eligible smokers for screening) 9
Mean number of LDCT screening centers per state in 2017	34 48
Percentage of adults without lung cancer screening center within 30 miles (persons aged 55 to 79 in 2017)	14.9% 48
Percentage of adults without lung cancer screening center within 30 min drive (persons aged 55 to 79 in 2017)	28.1% 48
Percentage of urban residents with access to LDCT screening center within 30 miles in 2017	93.7% 48
Percentage of urban residents with access to LDCT screening center within 30 min drive in 2017	83.2% 48
Percentage of rural residents with access to LDCT screening center within 30 miles in 2017	47.5% 48
Percentage of rural residents with access to LDCT screening center within 30 min drive in 2017	22.2%

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
