# Peer review of "Assessment of Barriers and Challenges to Screening, Diagnosis, and Biomarker Testing in Early-Stage Lung Cancer"

_cancers, 2023, doi:10.3390/cancers15051595_

Round 1
Reviewer 1 Report
The authors evaluated the potential barriers to low-dose computed tomography (LDCT) screening and diagnosis as well as to explore biomarkers and molecular testing in early-stage non-small cell lung cancer (NSCLC). This narrative review would help us to understand current developments in diagnosis, biomarkers, and molecular testing in early-stage NSCLC. This manuscript is logical and interesting. However, the following points should be addressed.
1. Artificial intelligence (AI) can potentially increase the efficiency of LDCT screening.
2. Is there another image screening such as FAPI-PET? Please describe new technologies for lung cancer screening and diagnosis.
3. How can we combine the image screening and biomarker/molecular testing in future? Effective biomarkers or evidence-based image scans used in combination may provide the clinician with effective tools for the early detection of NSCLC in the era of molecular and digital oncology. Please describe future perspectives.
4. Cost-effectiveness balance of the new tests should be discussed.
5. Please check “Simple summary”.
Author Response
Reviewer 1
The authors evaluated the potential barriers to low-dose computed tomography (LDCT) screening and diagnosis as well as to explore biomarkers and molecular testing in early-stage non-small cell lung cancer (NSCLC). This narrative review would help us to understand current developments in diagnosis, biomarkers, and molecular testing in early-stage NSCLC. This manuscript is logical and interesting. However, the following points should be addressed.
Artificial intelligence (AI) can potentially increase the efficiency of LDCT screening.
This is a great point and we have elaborated on this throughout the manuscript.
Is there another image screening such as FAPI-PET? Please describe new technologies for lung cancer screening and diagnosis.
Thank you for the suggestions. We added this as well.
How can we combine the image screening and biomarker/molecular testing in future? Effective biomarkers or evidence-based image scans used in combination may provide the clinician with effective tools for the early detection of NSCLC in the era of molecular and digital oncology. Please describe future perspectives.
Thank you for this suggestion. We have incorporated it in the conclusion section.4. Cost-effectiveness balance of the new tests should be discussed.
Thank you for the comment. We have added this as well.
- Please check “Simple summary”.
We will check “simple summary.”
Reviewer 2 Report
Review of "Assessment of Barriers and Challenges to Screening, Diagnosis, and Biomarker Testing in Early-Stage Lung Cancer"
Overall, well written narrative review on low-dose CT (LDCT) lung cancer screening. The question of barriers to low-dose CT (LDCT) lung cancer screening is relevant.
Major:
An overview of previous reviews on low-dose CT (LDCT) lung cancer screening will improve the manuscript, and the authors must demonstrate how the current review distinguishes itself from them (see a few examples below):
1) Lung Cancer Screening: Review and 2021 Update (PMID: 35402145)
Screening for Lung Cancer With Low-Dose Computed Tomography: An
2) Updated Evidence Report and Systematic Review for the US Preventive Services Task Force (doi:10.1001/jama.2021.0377)
A table with all the clinical trials reviewed and their associated major conclusions will help the reader get an overview of the topic.
Mass spectrometry on liquid biopsies has also been explored for biomarker research in lung cancer and must be discussed in the section on "Future Advancements in Biomarker Technologies". Examples are:
Assessment of a Large-Scale Unbiased Malignant Pleural Effusion Proteomics Study of a Real-Life Cohort [PMID: 36139528]
Is the Proteome of Bronchoalveolar Lavage Extracellular Vesicles a Marker of Advanced Lung Cancer? [PMID: 33233545]
The current conclusion section reads more like a summary of covered tasks than a conclusion. Furthermore, the authors do not sufficiently address the implications of the review.
Minor:
Line 123 efforts should be made to educate patients on both the risks of benefits of LDCT?
Author Response
Reviewer 2
Review of "Assessment of Barriers and Challenges to Screening, Diagnosis, and Biomarker Testing in Early-Stage Lung Cancer"
Overall, well written narrative review on low-dose CT (LDCT) lung cancer screening. The question of barriers to low-dose CT (LDCT) lung cancer screening is relevant.
Major:
An overview of previous reviews on low-dose CT (LDCT) lung cancer screening will improve the manuscript, and the authors must demonstrate how the current review distinguishes itself from them (see a few examples below):
1) Lung Cancer Screening: Review and 2021 Update (PMID: 35402145)
Screening for Lung Cancer With Low-Dose Computed Tomography: An
Our review paper attempts to combine the current data and approach in the detection of lung cancer using not just LDCT, but also incorporating biomarker testing and other novel approaches. We have added additional comments to the paper to refine it.
2) Updated Evidence Report and Systematic Review for the US Preventive Services Task Force (doi:10.1001/jama.2021.0377)
Thank you for the comment. Our paper attempts to go more in depth with biomarker testing as the paper cited above and others are more focused understandably on LDCT.
A table with all the clinical trials reviewed and their associated major conclusions will help the reader get an overview of the topic.
We are a little unsure on what clinical trials you are referring to. Please clarify this comment and we can address. Thank you.
Mass spectrometry on liquid biopsies has also been explored for biomarker research in lung cancer and must be discussed in the section on "Future Advancements in Biomarker Technologies". Examples are:
Assessment of a Large-Scale Unbiased Malignant Pleural Effusion Proteomics Study of a Real-Life Cohort [PMID: 36139528]
We have added this to the manuscript. Thank you for the great comment.
Is the Proteome of Bronchoalveolar Lavage Extracellular Vesicles a Marker of Advanced Lung Cancer? [PMID: 33233545]
We have added this as well to the manuscript. Thank you for the comment.
The current conclusion section reads more like a summary of covered tasks than a conclusion. Furthermore, the authors do not sufficiently address the implications of the review.
We have modified/added to the conclusion to address the implications of the review.
Minor:
Line 123 efforts should be made to educate patients on both the risks of benefits of LDCT?
Thank you for catching this. We have made the change.
Reviewer 3 Report
This manuscript reviews low-dose computed tomography (LDCT) screening and how this screening modality may be underutilized in early-stage non-small cell lung cancer (NSCLC). Current developments in diagnosis, biomarkers, and molecular testing in early-stage lung cancer are also evaluated. This is significant topic, which is of interest to readers. Mino weaknesses of the manuscript need to be improved by the authors for possible publication in Cancer.
Specific comments:
Line 114-117, “In the NLST trial, only 4% of participants were Black and yet 114 a secondary analysis suggested that Black individuals had a significant benefit from LDCT 115 screening. One approach to address disparities faced by African Americans and other 116 at-risk populations,……” Although “Black” and “African Americans” are the same, it is better to consistently use African Americans in the manuscript.
Line 133-141, “The upcoming FANS (Female Asian Never Smokers) Study will further explore potential causes of lung cancer in never-smoker Asian American women, which could in turn in-form screening strategies for never smokers with lung cancer.” Published papers reported that the concentrations of PM2.5, total carbon, and total nitrogen were higher during periods of cooking, especially for the fry and stir-fry methods, than during periods with non-cooking. Stir-frying is a cooking method, originating from Asia, in which food is fried in small amount of very hot oil. Cooking food with Stir fry method may be a risk factor since more Asian American women performing food than men. It is better to search such publications and add related information in the manuscript.
Author Response
Reviewer 3
This manuscript reviews low-dose computed tomography (LDCT) screening and how this screening modality may be underutilized in early-stage non-small cell lung cancer (NSCLC). Current developments in diagnosis, biomarkers, and molecular testing in early-stage lung cancer are also evaluated. This is significant topic, which is of interest to readers. Mino weaknesses of the manuscript need to be improved by the authors for possible publication in Cancer.
Specific comments:
Line 114-117, “In the NLST trial, only 4% of participants were Black and yet 114 a secondary analysis suggested that Black individuals had a significant benefit from LDCT 115 screening. One approach to address disparities faced by African Americans and other 116 at-risk populations,……” Although “Black” and “African Americans” are the same, it is better to consistently use African Americans in the manuscript.
We have changed this and use African Americans consistently.
Line 133-141, “The upcoming FANS (Female Asian Never Smokers) Study will further explore potential causes of lung cancer in never-smoker Asian American women, which could in turn in-form screening strategies for never smokers with lung cancer.” Published papers reported that the concentrations of PM2.5, total carbon, and total nitrogen were higher during periods of cooking, especially for the fry and stir-fry methods, than during periods with non-cooking. Stir-frying is a cooking method, originating from Asia, in which food is fried in small amount of very hot oil. Cooking food with Stir fry method may be a risk factor since more Asian American women performing food than men. It is better to search such publications and add related information in the manuscript.
We have added this as well. Thank you for the comment and suggestion.
Reviewer 4 Report
This review mainly describe the current assessment of barrier and challenges in LDCT screen for the diagnosis for early stage of lung cancer. The LDCT screen sample cohort is adult aged 50~80 year old who have 20 pack-year smokers which is limited to smoker population, but did not cover the non-smoker lung cancer populations which is really a limitation in outcome.
The manuscript is too descriptive public health outcome and no critical or significant aspects to inspire readers.
If the LDCT image data analysis out come from big image data machine learning and AI facilitates and reveals any significant foresights will be acceptable.
Author Response
Reviewer 4
Comments and Suggestions for Authors
This review mainly describe the current assessment of barrier and challenges in LDCT screen for the diagnosis for early stage of lung cancer. The LDCT screen sample cohort is adult aged 50~80 year old who have 20 pack-year smokers which is limited to smoker population, but did not cover the non-smoker lung cancer populations which is really a limitation in outcome.
We agree with this comment.
The manuscript is too descriptive public health outcome and no critical or significant aspects to inspire readers.
We have tried to go back and add a bit more to make it more novel and inspire readership.
If the LDCT image data analysis out come from big image data machine learning and AI facilitates and reveals any significant foresights will be acceptable.
We have added some future direction in our paper on how machine learning/AI can be incorporated.
Round 2
Reviewer 2 Report
The authors have improved the manuscript in accordance with the reviewers' suggestions. I only have one minor comment.
Minor:
L. 202 "Fibroblast activation protein inhibitor (FAP)," vs L.204 The fibroblast activation protein inhibitor (FAPI)
Author Response
We have made this correction. Thank you
Reviewer 4 Report
Still too descriptive and no significant point of views to readers.
If authors can specifically give the LC-MS Biomarkers and AI facilitated images outcome correlations results which potentially enlightening this manuscript.
Author Response
Still too descriptive and no significant point of views to readers.
We have gone back and made more specific points and removed some of the descriptive components.
If authors can specifically give the LC-MS Biomarkers and AI facilitated images outcome correlations results which potentially enlightening this manuscript.
We have added this to the paper. Thank you for the suggestion.